# Milk and Meat Allergens from *Bos taurus* β-Lactoglobulin, α-Casein, and Bovine Serum Albumin: An In-Vivo Study of the Immune Response in Mice

**DOI:** 10.3390/nu11092095

**Published:** 2019-09-04

**Authors:** Ewa Fuc, Dagmara Złotkowska, Barbara Wróblewska

**Affiliations:** Department of Immunology and Food Microbiology, Institute of Animal Reproduction and Food Research, Polish Academy of Sciences in Olsztyn, Tuwima 10 Str., 10-748 Olsztyn, Poland

**Keywords:** cross-reactivity, immune response, mice allergy model, cow milk, beef

## Abstract

The mechanism of food allergy may vary. This study aimed to compare the effects of milk, yogurt, or beef meat supplementation on humoral and cellular immune responses in a mice model. Mice were divided into four groups: The “Milk group” was sensitized with a β-lactoglobulin (β-lg)/α-casein (α-CN) mixture and supplemented cow milk; the “Yogurt group” was sensitized with β-lg/α-CN and supplemented yogurt; the “Beef group” was immunized with bovine serum albumin (BSA) and supplemented beef meat; and the “PBS group” received PBS in all procedures. ELISA was used to measure humoral response, including: Total IgE, specific IgG, and IgA. Cellular response was determined by phenotyping lymphocyte from lymphoid tissue and measuring the Th1/Th2 cytokine concentration with flow cytometry. The qPCR method was used for quantification of the fecal microbiota. The results obtained revealed a lower IgE level for the Yogurt group than for the Milk one. In the Yogurt group, the contribution of regulatory T cells to MLN and PP was higher compared to the other groups. We confirmed that diet supplementation with yogurt modulates the immune response to the prime allergen, and changes the activity of serum antibodies to milk proteins and BSA. Based on a specific antibodies level, we cannot exclude the possibility of CMA mice reaction against BSA.

## 1. Introduction

Food allergy is the most prevalent in infants and children, and affects as many as 8% of children and 2% of adults in Western countries [1]. The most common type of food hypersensitivity occurring in early childhood is cow’s milk protein allergy (CMA), which affects approximately 2.5% of children. CMA is associated with an increased risk of developing other allergic diseases, such as atopic eczema or allergic asthma [2]. The most effective approach to controlling CMA is complete avoidance of allergens. However, the diet in infancy and early childhood is often based on cow’s milk. Removal of milk from the diet would deprive children of a source of complete proteins rich in essential amino acids, branched-chain amino acids, vitamins D and E, and macronutrients such as iron and calcium [3,4]. There are certain technological processes to reduce the allergenicity of proteins, e.g., thermal denaturation and glycation [5,6]. Fermentation by proteolytic enzymes from Lactobacillus destroys milk protein epitopes and thus reduces protein allergenicity [7,8].

Food allergy is an abnormal immunological reaction associated with the digestion and absorption of food proteins, in which specific immunoglobulin (Ig)E antibodies are induced. The recognition of epitopes determines the direction of the immune system response towards hypersensitivity or tolerance. The allergic response is a result of immune processes mediated by Th2 cells. Once activated, these lymphocytes secrete cytokines, including IL-4, which results in immunoglobulin switching from class B to E and consequently, the activation of mast cells and development of allergic inflammation. CD4 T cells are a heterogeneous population of immune cells involved in immune response development. CD4 cells can differentiate into the Th2 subtype and play a major role in the induction of humoral responses and allergic diseases [9]. Their subpopulation, CD4^+^CD25^+^FoxP3^+^ T cells (regulatory T cells, Tregs), is involved in regulatory processes, including protection against inflammation and induction of peripheral tolerance [10]. CD8 T lymphocytes are also involved in IgE-mediated allergy. Their role is not well understood; however, as a source of IL-13, they reportedly exacerbate allergic symptoms [11].

There are 11 classified bovine milk allergens on the official World Health Organization/International Union of Immunological Societies list [12]. Caseins (Bos d8) and β-lactoglobulin (β-lg; Bos d5) are the strongest allergens, but the list also includes bovine serum albumin (BSA; Bos d 6). Caseins comprise nearly 80% of total milk proteins, and β-lg and BSA nearly 3% each [12,13,14]. Cross-reactivity between BSA and other bovine proteins has been observed in milk and meat, suggesting that BSA is a potential allergen for CMA patients [15]. However, BSA is considered a minor allergen in CMA [16], whereas it is one of the most important allergens in beef allergy. In children with meat hypersensitivity, BSA was found to be the most frequent protein involved in IgE-binding, as indicated by skin prick tests [17]. Beef allergy occurs in 3.28–6.52% of children with atopic eczema, whereas it is estimated to occur in 0.3% of the general population [16,17,18,19,20]. Chruszcz et al. [19] showed that patients with CMA react to serum albumin from meat of a variety of mammals, which is associated with a higher risk of clinical symptoms. In children with beef allergy, clinical reactivity with milk proteins has also been found [19]. Sensitivity to BSA in children with beef allergy could be the leading predictive marker for CMA detection [20].

The present study evaluated the effects of dietary supplementation with proteins from milk as a reference source of allergens, yogurt as a lactic acid fermentation product with immunomodulatory properties, and beef meat as one of the most allergenic meats, on the immune response in Balb/CCmdb mice sensitized with β-lg and α-CN or BSA from *Bos taurus* (domestic cow). We hypothesized that BSA is a minor milk allergen and that dietary supplementation with dairy products can modify the humoral response to BSA in CMA mice, whereas BSA as major a beef meat allergen can change humoral and cellular immune system responses to main milk allergens.

## 2. Materials and Methods

### 2.1. Mouse Diet Suplementation Products

Milk (pasteurized, 2% fat, 30 g/L protein) and yogurt (3% fat, 39 g/L protein) were purchased from the local market and were lyophilized in an FD-8-55 freeze dryer (Heto-Holten, Shanghai, China). Beef meat was purchased from local farmers at the local market. The meat was cooked in water in a metal pot for 15 min, cooled down, cut to small pieces, and lyophilized. The protein content of all products was determined by the Kjeldahl method [21,22]. Lyophilized materials with 3 mg of protein were dissolved in 100 µL of phosphate-buffered saline (10 mM PBS, pH 7.4) and used to feed the mice.

### 2.2. Animals

Female Balb/CCmdb mice (8 weeks old, 17–23 g) were obtained from the Center of Experimental Medicine in Białystok, Poland. The animals were housed individually in ventilated cages at the Animal Facility of Institute of Animal Reproduction and Food Research, PAS in Olsztyn, Poland. Water and standard diet (TPF, Altromin, Germany; free of milk proteins and BSA) were provided ad libitum. The mice were acclimatized before experiments. The local ethics committee in Olsztyn approved all procedures (43/2015).

### 2.3. Experimental Protocol

Mice were allocated to four groups: Milk, Yogurt, Beef, and control PBS, (*n* = 10 mice/group; Figure 1). Mice in the Beef group were sensitized with BSA (100 μg/mouse; Sigma-Aldrich) dissolved in PBS. Mice in the Milk and Yogurt groups were sensitized thrice at weekly intervals by intraperitoneal (i.p.) injection of a mixture of α-CN and β-lg in PBS in the presence of Freund’s adjuvant (100 μg of proteins/mouse, at a 4:1 ratio; Sigma-Aldrich, Poznan, Poland). Mice in the Beef, Milk, and Yogurt groups were administered 3 mg beef, milk, or yogurt, respectively, for 14 consecutive days using a pipette. Mice in the PBS control group were administered PBS following the same schedule. On days 15, 22, and 29 of the experiment, the mice from Milk, Yogurt, and Beef groups were given 10 μg cholera toxin as a mucosal adjuvant, together with supplemented food. Blood and fecal samples were collected every week, starting from the 14th day of the experiment. Coagulated blood was centrifuged in a 5418R centrifuge (Eppendorf, Hamburg, Germany) at 16,900× *g* for 10 min. Fecal pellets were extracted with PBS containing 0.01% NaN_3_ at 4 °C for 20 min and centrifuged as described above [23]. The fecal extracts and serum samples were stored at −20 °C until analysis.

### 2.4. Isolation of Lymphocytes

Lymphocytes from spleens (SPL), mesenteric lymph nodes (MLN), and Payer’s patches (PP) were isolated according to standard protocols used in our laboratory [24]. Briefly, tissues were homogenized in RPMI 1640 medium supplemented with 10 mM HEPES and 10 units/mL penicillin-streptomycin solution (incomplete medium), and the resulting cell suspension was filtered through an 80-μm nylon filter. The splenocytes were incubated with red cell lysis buffer (Sigma) for 5 min to remove erythrocytes. The lymphocytes were washed with incomplete medium and centrifuged at 400× *g* at 10 °C for 10 min. The pelleted cells were resuspended in 1 mL of incomplete medium. After trypan blue (Sigma) staining, lymphocytes were counted using a Bruker cell counter.

### 2.5. Splenocyte Culture and Cytokine Measurement

Splenocytes (1 × 10^6^) were cultured in 96-well plates in RPMI 1640 medium containing 10% heat-inactivated fetal bovine serum, 1 mM non-essential amino acids, 1 mM sodium pyruvate, 1 mM HEPES, and 10 units/mL penicillin-streptomycin (complete medium). Splenocytes were stimulated with 100 μg/mL of antigen. Concanavalin A (10 μg/mL) was used as a positive control. Spleen cells were incubated at 37 °C under 5% CO_2_ for 120 h. Then, the cells were centrifuged at 400× *g* and the supernatants were collected and stored at −80 °C until analysis. The concentrations of IL-2, IL-4, IL-6, IL-17A, TNF, and IFN-γ were estimated on a BD LSR Fortessa Cell Analyzer using a BD Cytometric Bead Array Mouse Inflammation Kit (560485; BD Biosciences, San Jose, CA, USA), according to the manufacturer’s recommendation. Final concentrations were calculated with the FCAP Array 3.0 software (BD Biosciences, San Jose, CA, USA).

### 2.6. Lymphocyte Phenotyping

Lymphocytes were suspended in 200 μL of FACS buffer (PBS with 5% FBS) in FACS tubes. To each tube, a mixture of monoclonal antibodies, including APC Cy7 rat anti-mouse CD4 (552051; BD Biosciences, San Diego, CA, USA), FITC rat anti-mouse CD25 (553071; BD Biosciences, San Diego, CA, USA), PE rat anti-mouse CD3 (555275; BD Biosciences, San Diego, CA, USA), and Alexa Fluor 700 rat anti-mouse CD8a (557959; BD Biosciences, San Diego, CA, USA), was added, and the cells were incubated at 4 °C for 15 min. Then, the cells were washed with FACS buffer and fixed with 2% paraformaldehyde. For intracellular staining, after washing, cells were permeabilized with ice-cold methanol at room temperature for 20 min. After washing, the cells were incubated at 4 °C for 20 min with Alexa Fluor 647 rat anti-mouse FoxP3 (560401; BD Biosciences, San Diego, CA, USA). After washing, the stained cells were analyzed on BD LSR Fortessa Cell Analyzer equipped with DIVA software. Cells were gated as follows: The lymphocyte population was gated on the FSC/SSC channel and was further gated for the CD3^+^ population, which was then further gated into CD4 and CD8 populations. From the CD3^+^CD4^+^ cells, the CD25^+^ population was taken as a parent for FoxP3 assignment. Each sample was prepared in triplicate, and 50,000 events were collected.

### 2.7. Antibody Measurements

Total IgE was determined by sandwich ELISA using a mouse IgE ELISA kit (157718; Abcam, Cambridge, UK) following the manufacturer’s protocol. The test range was 6.25–400 ng/mL, with a sensitivity of 1.83 ng/mL.

Specific antibody titers were determined by indirect ELISA. Briefly, 96-well plates were coated with an antigen solution (10 μg/mL in PBS) and incubated at 37 °C for 1.5 h. Non-specific protein-binding sites were blocked with 1.5% gelatin in PBS (10 mM, pH = 7.4). After incubation, the plates were washed three times with PBS containing 0.5% Tween 20. Serial dilutions of sera or fecal extracts (in 50 μL) were added to the plates, which were then incubated at 37 °C for 1.5 h. After washing, the plates were incubated with anti-mouse HRP-labeled secondary antibody (Sigma) for 1 h. After washing, the peroxidase substrate ABTS (Millipore, Temecula, CA, USA) was added, and after a 1-h incubation at room temperature, the absorbance at 405 nm was measured on a Jupiter UVM spectrophotometer (Assys Hitech GmbH, Eugendorf, Austria). The endpoint titer (Ept) was expressed as the reciprocal dilution of the last sample dilution of 0.1 OD above the negative control.

### 2.8. Quantification of Fecal Microbiota by qPCR

One hundred milligrams of feces was used for bacterial DNA isolation with a Stool DNA Purification Kit (Eurx, Gdańsk, Poland), using a protocol with a bead-beating step on a FastPrep instrument (MP Biomedicals, Solon, OH, USA). Major bacterial groups (*Clostridium coccoides*, *Clostridium leptum, Bacteroides-Prevotella-Porphyromonas*) and genera (*Bifidobacterium Lactobacillus*) were quantified as described previously [25,26], with some modifications. In brief, 1 µL of 100× diluted DNA was added to a reaction mixture containing 10 µL of SYBR Green Jump-Start Taq Ready Mix (Sigma, Poznań, Poland), 1 µL of each primer (Appendix A), 2.0–5.0 mM of MgCl_2_ (Appendix A), and molecular biology-grade water up to 20 µL. PCRs were run in duplicate on a QuantStudio 6 System (Thermo Fisher Scientific/Life Technologies, Warszawa, Poland) using a thermal cycler program of polymerase activation (95 °C, 3 min) and 40 cycles of denaturation (92 °C, 15 s), primer annealing (Appendix A, 30 s), and elongation (72 °C, 30 s). Melting curve analysis was conducted to confirm the specificity of amplicons. A standard curve was constructed as described in detail by Fotschki et al. [25]. The data were analyzed with QuantStudio™ Real-Time PCR software (Thermo Fisher Scientific), normalized to the weight of the sample and DNA dilution, and expressed as log10 of the cell number per gram sample wet weight.

### 2.9. Statistical Analysis

Data are presented as the mean ± standard deviation (SD). One-way analysis of variance (ANOVA) and Tukey post-hoc tests were used to analyze data with a normal distribution, and in other instances, the Kruskal–Wallis test was used. *p*-values <0.05 were considered significant. Statistica software (StatSoft Corp., Kraków, Poland) was used for statistical analyses.

## 3. Results

### 3.1. Yogurt Suplementation Decreases Humoral Response

Serum levels of specific immunoglobulins of classes G and A against α-CN, β-lg, and BSA, and total IgE, which characterize the humoral response to the antigens, are presented in Figure 2, Figure 3 and Figure 4. We observed dynamic changes in the total IgE level after 14 and 21 days of the experiment.

Serum IgE levels in the experimental groups on days 14 and 21 varied widely between individuals (large SD values). The immune response stabilized in the next 9 days (day 30 of the experiment) (Figure 2). The highest IgE level was detected in the Milk group (111.7 ± 20.4 ng/mL). Feeding mice with yogurt resulted in a decrease in the IgE level to 45.9 ± 11.3 ng/mL (*p* < 0.001). Mice in the BSA group had a similar total IgE level (70.9 ± 18.4 ng/mL), which was significantly different from the levels in the Milk (*p* < 0.05) and PBS (*p* < 0.001) groups.

Immunization with α-CN/β-lg and feeding mice with milk or yogurt resulted in similar titers of anti-α-CN IgG (2^15.8±0.45^, 2^15.6±0.89^) and anti-β-lg IgG (2^17.2±0.45^, 2^16.4±0.55^) (Figure 3A,B). Mice in the Beef group exhibited similar titers of anti-β-lg and anti-α-CN IgG, which were nearly two times lower than those in the Milk and Yogurt groups (*p* < 0.001). Titer of the anti-BSA IgG (Figure 3C) in the Milk and Yogurt groups were 2^9.4±0.89^ and 2^9.8±0.83^, respectively, and were significantly lower from those in the PBS. Its titers were almost two times lower then specific IgG Ept in Beef group (*p* < 0.001).

Specific secretory IgA was determined in fecal samples collected on the final day (30 d) of the experiment (Figure 4). sIgA titers (~2^6^) in the Beef group were the highest (*p* < 0.001) for all antigens tested (Figure 4A–C). sIgA titers in the Milk and Yogurt groups were similar for all antigens tested, and were significantly different from those in the PBS group for anti-α-CN (2^4.2±0.45^ and 2^3.8±0.45^ vs. 2^2.2±0.45^; *p* < 0.001).

### 3.2. Cellular Immune Responses

#### 3.2.1. T Lymphocyte Profile in Lymphoid Tissues

The contribution of T lymphocyte subpopulations in MLN and PP was determined, and data are presented in Figure 5. T cell populations were assigned among the lymphocyte population. In each experimental group, the percentage of CD3^+^CD4^+^ T cells in MLN was approximately two times lower than that in the PBS group (*p* < 0.001; Figure 5A).

The lowest percentage of CD3^+^CD4^+^ cells (19.1 ± 0.1%) was found in the Yogurt group, but it was not significantly different from those in the Milk (24.6 ± 4.2%) and Beef (24.7 ± 3.9%) groups (both *p* < 0.001). There were no differences in the percentage of CD3^+^CD8^+^ T cells in MLN. Significant differences were noticed in the CD4^+^CD25^+^ T cell population; in the Yogurt group, the CD4^+^CD25^+^ population (8.65% ± 2.55%) was significantly larger than those in the Milk and (2.65 ± 0.85%, *p* < 0.01) and Beef (3.1 ± 0.1%, *p* < 0.01) groups. T cell induction resulted in an increase in the Treg population in the Yogurt group to 90.3 ± 4.1%, as compared to 55.6 ± 6.9 % in the Milk group and 81.95 ± 5.15% in the Beef group (*p* < 0.001; Figure 5B).

T lymphocyte profiles for the PP are presented in Figure 5C. The highest percentage of CD3^+^CD4^+^ T cells was found in the Yogurt group (27.8 ± 4.7%), as compared to 15 ± 4.4% in the Milk group (*p* < 0.01) and 23.65 ± 1.85% in the Beef group (*p* < 0.01). The percentages of CD4^+^CD25^+^ T cells in the Milk (1.75%), Yogurt (1.6%), and Beef (1.45%) groups were significantly lower than that in the PBS group (5.95%, all *p* < 0.01). The contribution of CD4^+^CD25^+^Foxp3^+^ was significantly different (*p* < 0.001) between all groups (Figure 5D). Similar to the findings in MLN, in PP, yogurt supplementation resulted in an elevated Treg level (97.25%) when compared to that in the Milk group (55.6%, *p* < 0.001). Interestingly, similar to the IgE concentration, the percentage of FoxP3^+^ T cells in the Yogurt group was similar to that in the Beef group, suggesting some similarities in induced immune mechanisms.

To clearly define trends in Treg induction, splenocytes from all experimental groups were cultured and stimulated with pure antigens. Treg activity differed between the groups (Figure 6). Cells obtained from mice of the Yogurt group had the highest CD4^+^CD25^+^Foxp3^+^ fraction, regardless of which antigen was used for stimulation. The lowest level of Tregs was found in the Beef group T cells. CD4^+^ cells obtained from mice of the Milk group actively responded to antigen stimulation.

Cultures stimulated with α-CN showed a significantly lower contribution of Tregs (27.2%; *p* < 0.01) than cultures stimulated with BSA (53.3%; *p* < 0.01) or growth in medium (56.0%; *p* < 0.01). A similar trend was observed upon β-lg stimulation, where the Treg percentage (22.9%) was almost two times lower than those upon BSA stimulation and in medium (*p* < 0.01). The stimulating antigen did not change the percentage of Treg in the Beef group cultures. This level did not exceed the level for the cells growing in the medium.

The results show that the three antigens from domestic cow evaluated in this study activate differential immune responses, as indicated by the differences in the contributions of CD4^+^, CD8^+^, and CD4^+^CD25^+^Foxp3^+^ among lymphocytes in the different experimental groups.

#### 3.2.2. In Vitro Cytokine Secretion

For a thorough characterization of the immune response, the secretion of IL-10, TNF-α, IL-6, and IL-4 cytokines from splenocytes stimulated with α-CN, β-lg, and BSA was measured (Figure 7A–D).

The Milk group splenocytes were very sensitive to α-CN and β-lg stimulation. After α-CN stimulation, the IL-10 level increased to 395.84 ± 42.61 pg/mL, in comparison to 114.92 ± 20.80 pg/mL after β-lg stimulation, 63.08 ± 7.89 pg/mL after BSA stimulation, and 57.31 ± 5.83 pg/mL in the absence of stimulation (*p* < 0.001, Figure 7A). Yogurt group splenocytes released similar levels of IL-10 after α-CN or β-lg stimulation (542.30 ± 56.02 pg/mL and 445.17 ± 117.88 pg/mL, respectively), which were 3–4 times higher than those in cells stimulated with BSA (108.09 ± 34.16 pg/mL) or grown in medium (96.7 ± 17.8 pg/mL, *p* < 0.001) These findings suggest that cells sensitized with BSA do not recognize milk protein in vitro. Beef group lymphocytes did not show a strong response in terms of cytokine secretion upon stimulation; the IL-10 level did not exceed those in non-stimulated cells and cells from the PBS group.

A significant increase in the TNF-α level (*p* < 0.05, *p* < 0.01) was seen after β-lg stimulation in each experimental group (Figure 7B). In the Yogurt group, statistically significant differences were observed after β-lg stimulation (243.23 ± 75.7 pg/mL) compared to α-CN stimulation (84.23 ± 7.9 pg/mL; *p* < 0.05) or BSA stimulation (66.55 ± 10.2 pg/mL; *p* < 0.01). In the Beef group, cells stimulated with β-lg released higher levels of TNFα (157 ± 37.4 pg/mL; *p* < 0.05) than cells stimulated with α-CN (28.84 ± 7.1 pg/mL, *p* < 0.05) or BSA (32.41 ± 6 pg/mL (*p* < 0.05). The Milk group cells released the highest amounts of IL-6 upon any stimulation (Figure 7C; α-CN, 788.52 ± 53.2 pg/mL, *p* < 0.001; β-lg, 1582.45 ± 85.9 pg/mL, *p* < 0.001; BSA, 1001.14 ± 26, *p* < 0.001) when compared to levels in cells from medium only. IL-4 secretion by splenocytes significantly responded to stimulation with different antigens only in the Yogurt group. The highest concentration was measured after stimulation with α-CN (30.3 ± 0.9 pg/mL), as compared to 21.66 ± 3.3 pg/mL after β-lg (*p* < 0.05) and 16.96 ± 2.8 pg/mL (*p* < 0.01) after BSA stimulation. Thus, the levels and profile of secreted cytokines was dependent on the antigen used for stimulation and the experimental treatment.

#### 3.2.3. Quantification of Colonic Microbiota

Quantification of the colonic microbiota revealed that the mice fed yogurt had higher levels of *C. coccoides* and *C. leptum*, and *Lactobacillus* (Figure 8). Statistically significant differences were observed between the groups fed yogurt and beef. The amount of *Bifidobacterium* differed significantly between the PBS and Yogurt (*p* < 0.05), Beef, and Milk (*p* < 0.001) groups. The highest amounts of all tested strains (except *Bifidobacterium*) were found in the Yogurt group.

## 4. Discussion

The aim of the present study was to test differences in the cellular and humoral immune responses of mice sensitized to three different allergens from domestic cow and fed milk, yoghurt, or beef. According to our knowledge, this is the first study to apply a multi-faceted approach to analyze the immune responses to different allergens from the same animal species, i.e., two major milk allergens, α-CN and β-lg, and the beef meat allergen BSA. Garcia et al. [27] reported that 10–20% of children with hypersensitivity to milk also experience hypersensitivity to cow meat. Additionally, 80–90% of children allergic to cow meat exhibit CMA. This study revealed differences in specific IgG responses between the experimental groups. Mice sensitized with milk proteins and fed dairy products had higher levels of anti-β-lg and anti-α-CN IgG than of anti-BSA IgG. Opinions on the role of IgG in the diagnosis of food allergy are divided. Gocki et al. [28] reported a few functions of specific IgG, one as a normal immune response to foreign antigen. Most often, attention is paid to the role attributed to relations between specific IgG and specific IgE antibodies. In humans, there are four subclasses of IgG, and it has been reported that a high level of specific IgG4 is effective in immunotherapy, owing to its protective or blocking role [29]. High-level specific IgG in children with IgE-mediated allergy is a predictor of future tolerance [28]. Specific IgG levels (Figure 3) were consistently higher in all experimental groups than in the PBS group, in line with findings in a previous study [30]. These indicated that mice developed a strong humoral response to the proteins used for immunization (Figure 2, Figure 3 and Figure 4). Anti β-lg and α-CN IgG and sIgA titers were comparable in the groups sensitized with milk proteins and fed milk or yogurt. Yogurt supplementation did not decrease specific IgG level compare to the Milk group, but fermented products are directed more for IgA secretion. BSA trated induced a variety of antibodies, which cross-reacted with α-CN and β-lg, and their titer is almost two times lower than for BSA (Figure 3). Maybe earlier introduction of BSA to the immune system could decrease the possibility of allergic reaction. This suggestion needs future study.

IgA produced in the mucous membrane is important for immune response [31]. Frossard et al. [32] reported that the level of secretory IgA was higher in tolerant mice and lower in allergic mice when IgE was used as a marker of food allergy. Interestingly, as for the fecal sIgA titer, mice fed beef homogenate had the highest levels of IgA against all antigens tested, and the levels were different from those in the PBS group (*p* < 0.05). This suggests that beef homogenate exerts a protective role in the intestinal mucosa. Anti-α-CN sIgA titers were significantly different between the Yogurt and Beef groups (*p* < 0.001; Figure 4). This fact presents α-CN as the strongest allergen from the three tested in this study. The BSA-treated group had almost 50% higher specific sIgA when compared to the Milk or Yogurt groups. This suggests that controlled diet supplementation with BSA could modulate immune system response to milk allergens. This hypothesis needs future study.

Significant differences in the IgE level were observed between groups on day 30 of the experiment (Figure 2). We observed a significantly reduced IgE level (*p* < 0.001) in mice fed yogurt. These results are in agreement with earlier findings [33], and show that fermented products such as yogurt reduce the specific IgE against sensitizing protein, but unfortunately do not reduce the risk of cross-reactivity. In the present study, feeding the animals with yogurt resulted in a long-term 50% reduction in the total IgE level when compared to the levels in mice supplemented with milk, and reduced the cross-reactivity to BSA by 50% when compared to the response to primary antigen.

In food allergy, it is important to characterize the cellular response in gut-associated lymphoid tissue, like PP and MLN, which are inductive sites and effector sites like SPL and HNLN. Multiple cells such as T lymphocytes, B lymphocytes, and macrophages respond to presented food antigens and induce immune system response. T cells and cytokines play a critical role in controlling allergic disease [34]. The present study showed that the contribution of Tregs among CD4^+^CD25^+^ of MLN and PP lymphocytes was lower in mice with diet supplemented with cow’s milk than yogurt (Figure 4B,D). For the BSA group, the percent of Tregs in PP was about 30% higher than in the Milk group (*p* < 0.001) and about 20% lower than in the Yogurt group (*p* < 0.001). In MLN, there were no differences in Tregs percentage between the Yogurt and BSA groups, but an increase was observed when compare to the Milk group (*p* < 0.01). Stimulation of cultured splenocytes derived from experimental mice resulted in differential contributions of Tregs among CD4^+^CD25^+^ T cells, depending on the group and antigen used (Figure 6). However, the percentage of Tregs among CD4^+^CD25^+^ T cells was the highest in the Yogurt group among all experimental groups (the same as in fresh cells presented on Figure 5), and no differences were observed among the antigens used for stimulation. The increase in the contribution of Tregs at inductive sites of gut-associated lymphoid tissue in mice from the Yogurt group suggests that fermented milk products could be used for faster acquisition of tolerance to cow proteins (Figure 5 and Figure 6). Tregs are very important immunoregulators in inflammatory states [35]. Noval et al. [36] and Kondělková et al. [37] reported that Tregs regulate hypersensitivity reactions in allergic disease by affecting mast cells and the IgE level. The amount of specific IgE was correlated with the percentage of Tregs in immunized mice fed horse milk [25]. In an animal model of oral tolerance, Mucida et al. [38] and Couper et al. [39] proved that tolerance was correlated with the induction of antigen-specific CD4^+^CD25^+^Foxp3^+^ Tregs.

An imbalance in Th1/Th2 and net cytokine release determines the development of a pro-inflammatory state or promotion of the allergic responses. Th2-type cytokines include ILs 4, 5, and 13, which are associated with the promotion of IgE. IL-10 cytokine is produced by Th1 cells and may affect the ability of other cells to produce IL-10 and control allergic inflammation [39]. The highest concentration of IL-10 was found in cultured splenocytes isolated from the Yogurt group mice and stimulated with β-lg or α-CN. This observation suggests that cells recognize the antigen encountered previously (during sensitization). The Beef group splenocytes do not release significant amounts if IL-6, IL-10, or IL-4 secretion after stimulation. The highest concentration of IL-4 we found in the Milk group when compared to the Yogurt and BSA groups. IL-4 strongly stimulates B lymphocytes and leads to switching to IgE, which is important in the pathomechanism of allergy, and was also observed in the presented experiment (Figure 2). It also affects T lymphocytes, directing their development towards Th2 cells. The experiments using splenocytes showed that stimulation with α-CN induces the highest level of TNF and IL-6 in each experimental group.

The microbiome plays an important role in the regulation of allergic diseases [40]. In an earlier study, the amount of *Lactobacilli* was found to be lower in allergic than in non-allergic children, based on a comparison of fecal microbiota [41]. In addition, Treg induction can change commensal microflora, including *Clostridium* [42]. In this study, we observed that the group supplemented with yogurt had the highest amounts of *C. leptum* and *C. coccoides* when compared to the other treatment groups, which can be attributed to the contribution of Tregs. At the same time, the lowest IgE concentration was noticed. The correlation between IgE level, Treg contribution, and commensal bacteria is visible, but deeper futures studies are needed.

In conclusion, this study confirms that CMA is not always associated with hypersensitivity to beef. We found that diet supplementation with yogurt reduces the allergenicity of milk, but does not reduce the risk of cross-reactivity between milk proteins and BSA. The present model may be useful for understanding the mechanisms underlying allergic reaction and developing new products to prevent food hypersensitivity, as well as to gain a deeper understanding of the correlation between humoral and cellular responses. Our results show that it is possible to silence the immune response by administering modified by fermentation allergenic proteins, with weak antigenic potential. Our in vitro results suggest that early exposure to BSA could also change immune system response to milk allergens, but this need more study. To fully characterize the mechanism of cross-reactivity between milk proteins and BSA, a variety of milk proteins should be studied in the future. Further, future studies should focus on the possibility of immune response induction by adoptive transfer of sensitized lymphocytes.

## Figures and Tables

**Figure 1 nutrients-11-02095-f001:**
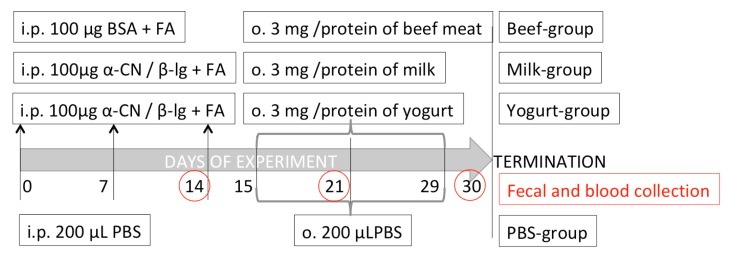
Experimental setup.

**Figure 2 nutrients-11-02095-f002:**
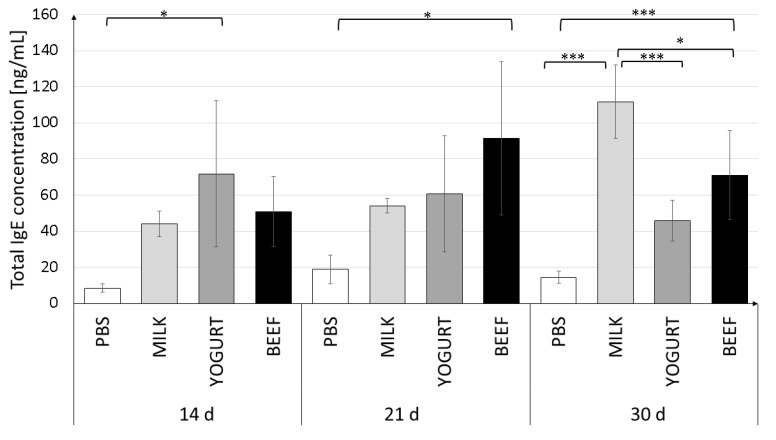
Total serum IgE levels in mice on days 14, 21, and 30 of the experiment. Mice were divided into four treatment groups (*n* = 10): Mice in the Milk and Yogurt groups were immunized with α-CN/β-lg and fed milk or yogurt, respectively. Mice in the Beef group were immunized with BSA and fed beef meat. Mice in the PBS group were administered PBS. Data are the mean ± SD. * *p* < 0.05, *** *p* < 0.001, one-way ANOVA with Tukey post-hoc tests.

**Figure 3 nutrients-11-02095-f003:**
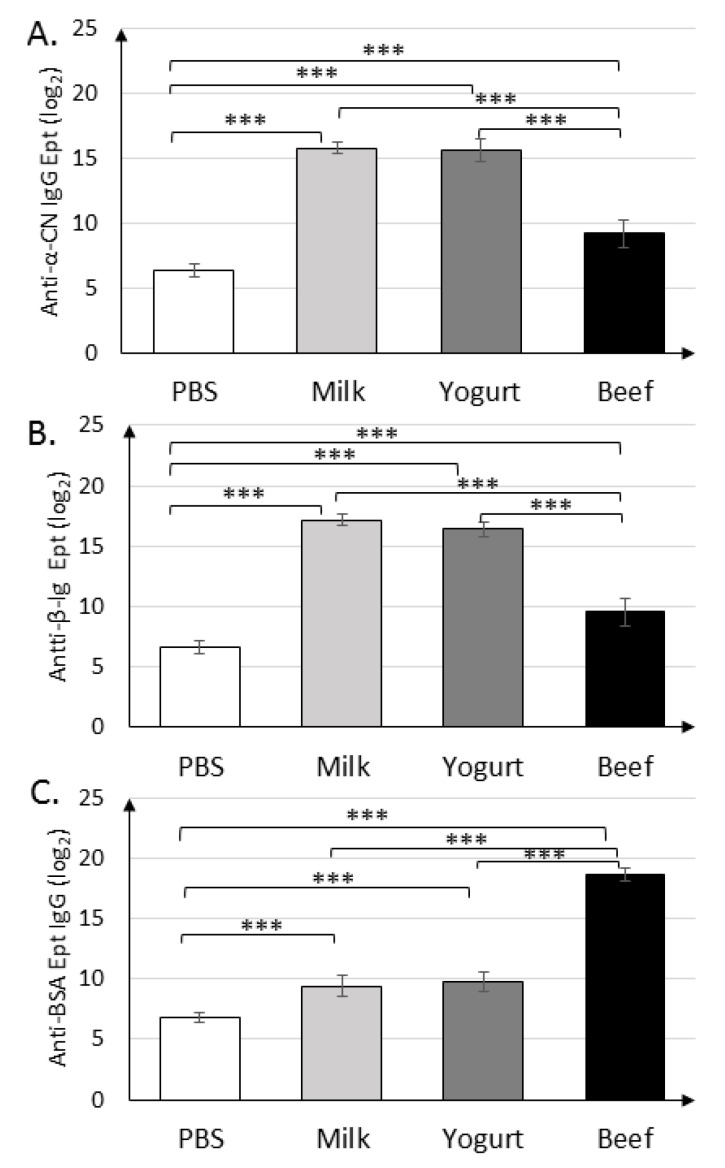
Serum levels of IgGs against α-CN (**A**), β-lg (**B**), and BSA (**C**) on day 30 in mice from the Milk, Yogurt, Beef, and PBS groups. Data are the mean (*n* = 10) ± SD, *** *p* < 0.001, one-way ANOVA with Tukey post-hoc tests.

**Figure 4 nutrients-11-02095-f004:**
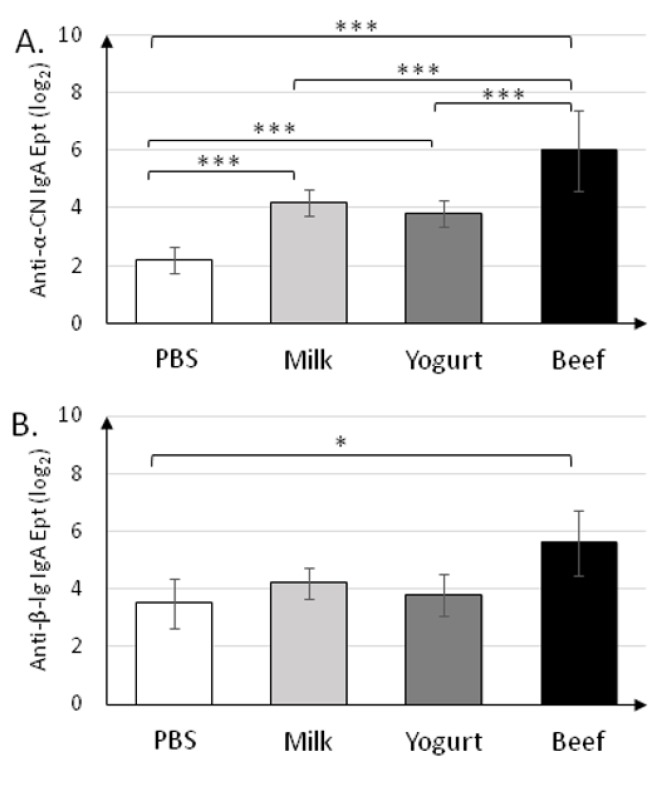
Levels of secretory IgA specific to anti-α-CN (**A**), anti-β-lg (**B**), and anti-BSA (**C**) in fecal samples collected on day 30 of the experiment. Data are the mean (*n* = 10) ± SD. * *p* < 0.05, ** *p* < 0.01, *** *p* < 0.001, Kruskal–Wallis test and one-way ANOVA with Tukey post-hoc tests.

**Figure 5 nutrients-11-02095-f005:**
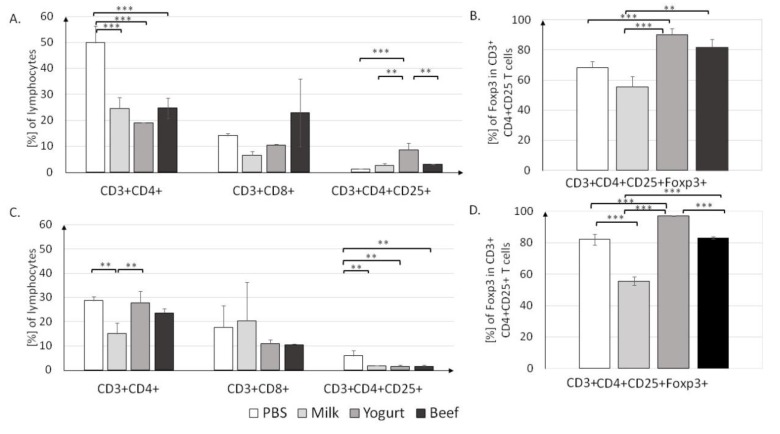
T cell profiles in the MLN (**A**,**B**) and PP (**C**,**D**) in mice from the PBS, Milk, Yogurt, and Beef groups. Cells were gated as follows: FSC/SSC lymphocytes were gated. Then, CD3^+^ and CD4^+^ or CD8^+^ cells were gated. From CD4^+^CD25^+^, FoxP3^+^ cells were gated. For each sample, 50,000 events were collected. Data are the mean ± SD. ** *p* < 0.01, *** *p* < 0.001, one-way ANOVA with Tukey post-hoc tests.

**Figure 6 nutrients-11-02095-f006:**
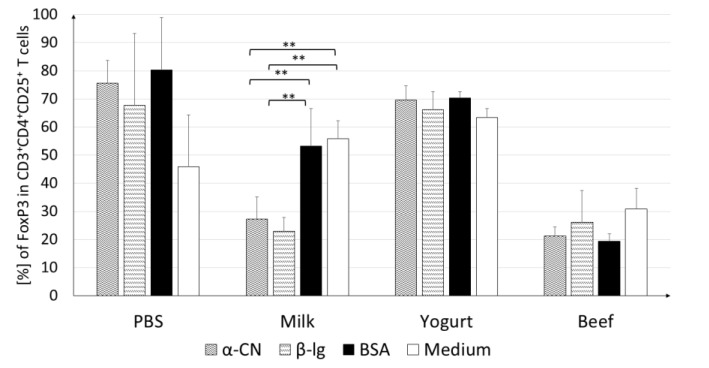
FoxP3^+^ cells among CD4^+^CD25^+^ splenocytes. Cells were stimulated with 100 µg/mL of α-CN, β-lg, or BSA and incubated at 37 °C for 120 h. Data are the mean ± SD. ** *p* < 0.01, one-way ANOVA with Tukey post-hoc tests.

**Figure 7 nutrients-11-02095-f007:**
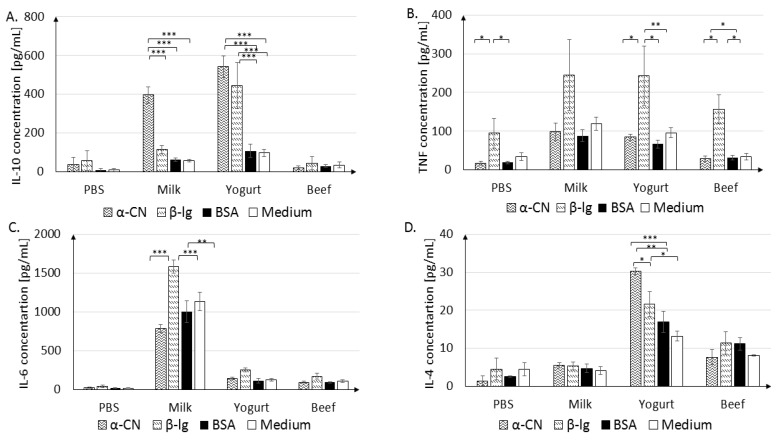
IL-10 (**A**), TNF (**B**), IL-6 (**C**), IL-4 (**D**) secretion in cultured splenocytes. Cells were stimulated with 100 µg/mL of α-CN, β-lg, or BSA, or grown in medium, and were incubated at 37 °C for 120 h. Data are the mean ± SD. * *p* < 0.05, ** *p* < 0.01, *** *p* < 0.001, one-way ANOVA with Tukey post-hoc tests.

**Figure 8 nutrients-11-02095-f008:**
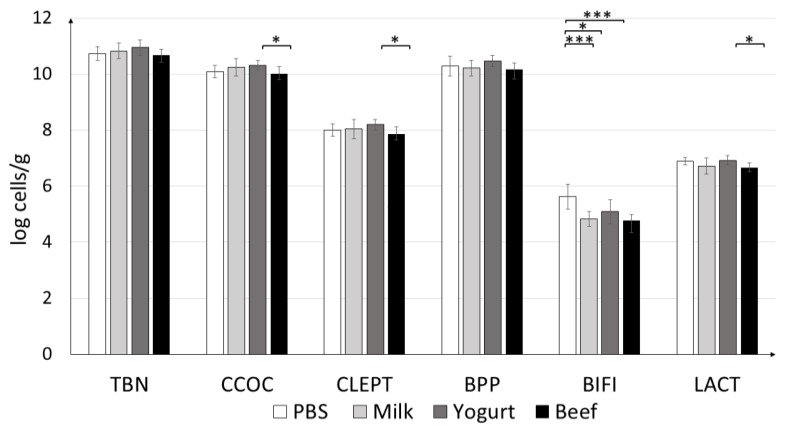
Effects of antigen treatments on mouse fecal microbiota profile. qPCR results are presented as log_10_ of bacterial cells per gram of the wet weight of the fecal content of experimental animals. Data are the mean ± SD. * *p* < 0.05, *** *p* < 0.001. TBN = total bacteria; CCOC = *Clostridium coccoides*; CLEPT = *Clostridium leptum*; BPP = *Bacteroides-Prevotella-Porphyromonas*; BIFI = *Bifidobacterium*; LACT = *Lactobacillus*.

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
