# Peer review of "Milk and Meat Allergens from Bos taurus β-Lactoglobulin, α-Casein, and Bovine Serum Albumin: An In-Vivo Study of the Immune Response in Mice"

_nutrients, 2019, doi:10.3390/nu11092095_

Round 1
Reviewer 1 Report
Please check the attached file.

Author Response
Thank you for all your comments and suggestions. We highlighted all changes in the manuscript. We refer to the comments below in detail and indicate where they are included in the manuscript.
On behalf of all the authors, I would likealso apologies for the English language grammar. Manuscript was corrected using professional service Ediatge Cactus (certificate included) and we did not expect this kind of feedback. We are very sorry for that. On that moment we have no enough time for second correction. We would be very grateful if Editor and Referee agree for review the manuscript as it stands now.
The introduction should include background information pertaining to the humoral and cellular response.
AU: You are rigt about this. We’ve placed the paragraph in lines 32-43.
This paper has quite a few grammatical errors and the essay structure is weak; it should be proofread by professionals.AU: AS I mention earlier we are very sorry for this situation. Before manuscript application to Nutrients we scheduked correction by professional service (Editage by Cactus). Right now we have no time to correct it by professional. We did some correction by ourselves and we understand it is not enough. We will schedule the service but we will be very grateful if you agree review manuscript in this form.
The different treatment groups should be clearly defined, and the same abbreviations should be consistently used to avoid lengthy and complicated descriptions.AU: We have completed the description of the experimental groups. Additionally we added Figure 1 with experimental schedule. We hope it will make the schedule of the experiment more clear.
The graphs are too small, and the graph titles should include descriptions of the subgraphs (A, B, C, D…etc.)AU: We corrected the graphs.
In line 310, the author mentioned, “T-cells and cytokines play a critical role in controlling allergic disease…” but did not elaborate on the similarities and distinctions between the T-lymphocyte profile of the present study and the published literatures.AU: Thank you for suggestion. The information about subpopulation of lymphocytes was added.
The author should explain why MLNs and splenocytes were used and how does that contribute to the differences in their T-lymphocyte profiles.AU: The explanation about the role of MLN in food allergy was added.
“An important in food allergy is to characterize the cellular response in GALT tissue. GALT tissue includes Payer’ Patches, mesenteric lymph node (MLN). The different part of lymphoid GALT contains immune cells such as: lymphocytes T, lymphocytes B, macrophages. Is known that, T cells and cytokines play a critical role in controlling allergic disease.”
Suggestions and revisions:
Line 13-15, elaborate on the abbreviation of CMA, b-lg, a-CN, BSA and PBSAU: The Journal do not recommend to add separate section for abbreviation, we add it in lines 419-427. If editor agree it will stay in manuscript. Anyway we explain them in first use. The meaning are listed below: CMA- cow milk allergy, α-CN- α-casein, β-lg- β-lactoglobulins, BSA- bovine serum albumins, PBS- phosphate buffered saline, MLN- mesenteric lymph nodes, PP –Payer’ patches , EpT- End point Titer
Line 14, beef meatAU: Thank you. I corrected it.
Line 20-21, what is the significance of the findings and what can you conclude? implications and future directions are missingAU:
Line 31, what are the valuable nutrients associated with CMA? be more specificAU: Probably the sentence was wrong. Of course CMA has no any nutrients. We were thinking about baby’s diet based on milk which is complete in proteins, fat, carbohydrates substrate used for baby’s formula preparing. We are very sorry for that. We corrected sentence.
Line 35, foundAU: It was corrected.
Line 43, what are some previous research being conducted on BSA and hypersensitivity? what is the knowledge gap that led to the research questionAU: Bovine serum albumins (BSA) is considered to be a minor allergen in CMA. However the content of BSA in milk is similar too b-lg the second after casein milk allergen. So it is interesting to compare the humoral and cellular immune response induce by BSA in comparioson with two main bovine milk allergens. BSA is present in milk and milk products. BSA is also from the same animal cattle as a-CN and b-lg. In case of beef allergy BSA is one of the most important allergens which is present in milk products not only the meat. In patients with meat hypersensitivity it was observed that BSA was the most frequent protein involved in IgE-binding. All children who took part in the study also showed a positive skin prick test (SPT) for BSA. We have completed the tekst.
Line 52-54, name of the local market? the brand and company product? perhaps better if you trace the source of beef to the farm?AU: We don’t have permission to disclose brand of the milk, yogurt, meat producer or name of the supermarket. However we know, that the source of those products were the local factory and farmers. We included this information.
Line 54, the cooking condition for beef should be more detailed.AU: The beef meat purchased at the local market was cooked in the boiling water in a metal pot for 15 minutes. Next was cooled, cut to small pieces, and lyophilized.
Line 130, cecal digestateAU: We corrected it: fecal
Line 167, were found. “milk” group.AU: We corrected it.
Line 168, it's inaccurate to use the phrase 'yogurt decreased the level' when it's an independent treatment group; better to say 'decreased the level induced by a-CN'.AU: We corrected it.
Line 170, foundAU: We corrected it.
Line 172 “sililar titer”?AU: We corrected it.
Line 173, milk and yogurt groupsAU: We corrected it.
Line 174, orf?AU: We corrected it.
Line 177, the figure is unclear, graphs should be bigger and re-arranged in 2/row instead of 3 in one column. Rearrange the lines with asterisks for more organized patterns.AU: The figures were improved. The axis name font has been enlarged and the quality of the graphs was improved.
Line 178, Figure 2. …and BSA “induced” IgG titers.AU: We corrected it.
Line 182, suggestive presentation: The specific secretory IgA (sIgA) of feces collected at day 30 was determined.AU: The title of Fig.3 was changed.
Line 196, “depictured”?AU: We changed it to presented.
Line 201, what does Figure 4 A, B, C, D mean? The division of A~D groups should be noted and explained in figure and legend.AU: We corrected the description of figure 4.
Line 212, an increase “of yogurt group” in Treg populationAU: We corrected it: Induction in T cells also resulted in an increase of Treg population in yogurt group (..)
Line 319, Mention the names of the researchers instead of 'other'. What do you mean by 'affecting mast cells'? Please elaborate how do the previous findings relate to your study?AU: Thank you for suggestion. We corrected it and we tried to compare our results.
Line 319, Treg (Figure 7).AU: Sorry for that. We added the Fig.7.
Line 328, 'fermented milk product' is too general and the results is not generalizable because only yogurt was tested; yogurt is not the only type of fermented milk product, will cheese induce a similar effect?
AU: Thank you for suggestion. In our experiment we used only one commercial yogurt. Based on the knowledge from our team work type fermented products depends from microorganism differently change milk protein immunity, and differently modulate immune response. We agree that we shoud precisely state only about product we used yogurt. Therefore we changed “fermented products” with yogurt.
Line 337, what are some other studies that corroborated your findings or investigated the underlying rationale?AU: We completed the text.
Line 343, future directions?AU: We included paragraph about future research.
However, in order to fully characterize the phenomenon of cross-reactivity protein found only in beef meat of only in milk should be use. The future studies should be focus on possibility induction of immune response by adoptive transfer sensitized lymphocytes, which would be increase knowledge about the phenomenon of cross-reactivity
Reviewer 2 Report
Overview:
The authors have conducted a mouse experiment in which mice are sensitized with milk proteins (alpha-casein, beta-lactoglobin, or albumin) and then fed milk, yogurt, beef, or PBS. The humoral and cellular responses are then characterized as well as some limited aspects of the microbiome. Overall,the results presented in the figures are quite interesting and would advance the field, but the hypotheses are not stated, the results are poorly conveyed (whatare themain results?), and there is inadequate discussion of the results.
Major comments:
1) What is the main hypothesis of the paper? It is difficult for the reader to know what hypotheses are being tested as none are stated.
2) The introduction should explain the necessary background and lead to the limitations of prior work and why the current work is needed. What led the authors to do this experiment?
3) Albumin is not simply a beef protein. It is a major protein in bovine milk! This should be disclosed in the paper.
4) If the authors wanted to test a hypothesis about cross-reactivity, wouldn't they choose a beef protein that is not also a major protein in milk?
5) The experiment is incompletely explained. What was the feeding regimen? Form and dose of the milk, yogurt, and beef? What processing was applied to the milk? (the type of heat treatment is known to affect allergenicity). Was the meat fully cooked? How were the mice fed these foods?
6) The major findings are difficult to discern because they are not plainly stated in the text.
7) Has cross-reactivity been shown? There should be a paragraph or two in the Discussion that presents all of the evidence for (or against) the case for cross-reactivity from the experiments and in the context of existing literature.
8) It seems like the most interesting aspects of the study are the milk-yogurt comparisons. There should be a paragraph in the Discussion section that summarizes the milk and yogurt differences and what are the possible mechanisms that may be driving these differences.
9) In general, the Discussion section is poorly written...both in content and style. There is a lack of logical flow. There are two sentence paragraphs and incomplete thoughts.
10) The Discussion should include a paragraph on limitations of the study. Some obvious limitations: use of BSA to test cross-reactivity when it is already a milk protein. Use of a few primers to probe the gut microbiome. What should be studied in the future?
11) The Discussion section begins with "This research aimed to show..." Research should never aim to show something. Research should test hypotheses.
Minor comments:
1) The authors need to be clear about what fermentation does and does not do. In the Introduction, there is a statement that "lactate fermentation naturally destroys the epitopes of milk proteins". How?? What is the mechanism? What paper demonstrates this? A sentence or two of explanation would help.
2) Abbreviations are used in the abstract before they are defined.
3) All figures are much too small with some text completely unreadable.
4) There are spelling and grammar mistakes throughout the manuscript.
5) All figure legends need to explain the parts. If there is an A, B, C, D in a composite figure, then the legend needs to explain what is A. B. C. and D.
6) All y-axis and x-axis in figures need to be labelled appropriately. For example, Figure 1 y-axis is "concentration". Concentration of what ?
7) The units always need to be given. For example, the sentence ending in line 183 is missing the units.
8) Stylistically, it would be helpful to end every results section with some kind of summary sentence so that the reader knows what is the main finding of that section.
9) Stylistically, it would be helpful to begin each results section with a sentence that either states the hypothesis to be tested or gives the background of why the experiment was done.
10) The flow cytommetry experiments are lacking details. For example, it would be impossible to re-produce Figure 5 from what is presented in the Methods section.
11) Some suggested reading:
Bovine Milk Allergens: A Comprehensive Review
https://onlinelibrary.wiley.com/doi/pdf/10.1111/1541-4337.12318
Author Response
Thank you for all your comments and suggestions. We highlighted all changes in the manuscript. We refer to the comments below in detail and indicate where they are included in the manuscript.
On behalf of all the authors, I would likealso apologies for the English language grammar. Manuscript was corrected using professional service Ediatge Cactus (certificate included) and we did not expect this kind of feedback. We are very sorry for that. On that moment we have no enough time for second correction. We would be very grateful if Editor and Referee agree for review the manuscript as it stands now.
Major comments:
What is the main hypothesis of the paper? It is difficult for the reader to know what hypotheses are being tested as none are stated.AU: We evaluated the effect of diet supplementation with milk, yogurt and beef meat on the immune response of mice sensitized with β - lactoglobulin and α - casein or bovine serum albumin. We would like to check if BSA sensitized mice will respond to milk allergens. We made changes to manuscript try clarify our point.
The introduction should explain the necessary background and lead to the limitations of prior work and why the current work is needed. What led the authors to do this experiment?AU: We changed introduction. We hope you will find it properly did.
Albumin is not simply a beef protein. It is a major protein in bovine milk! This should be disclosed in the paper.AU: We agree with this suggestion. Bovine serum albumins is protein occurs in milk, however, this is small amounts 0.1-0.4g/L. BSA is considered to be a minor allergen in the case of CMA. The opposite situation occurs in the case of beef allergy, where BSA in considered to be the main allergen. In introduction we added paragraph explening.
Bovine serum albumins (BSA) is considered to be a minor allergen in CMA [Hochwalner 2014]. However in case of beef allergy BSA is one of the most important allergens. In patients with meat hypersensitivity it was observed that BSA was the most frequent protein involved in IgE-binding. All children who took part in the study also showed a positive skin prick test (SPT) for BSA [Restani et al. 1997].
If the authors wanted to test a hypothesis about cross-reactivity, wouldn't they choose a beef protein that is not also a major protein in milk?AU: We chose BSA, because this protein is the major allergens in beef meat allergy. We agree that the best solution would be to select the protein found in beef meat. Therefore, it should be tested in future studies.
The experiment is incompletely explained. What was the feeding regimen? Form and dose of the milk, yogurt, and beef? What processing was applied to the milk? (the type of heat treatment is known to affect allergenicity). Was the meat fully cooked? How were the mice fed these foods?AU: The milk, yogurt and boiled beef were lyophilized. The protein content of all product was determined by Kjedahl method. Lyophilized milk, yogurt, beef meat which contains 3 mg of protein were dissolved in 100ul PBS. In our research, the used products as the normally eat. That’s why the beef meat was cooked. The milk was not-heat treated. The description about preparing of sample feeding is in paragraphs:
We complete the point 2.2 Commercial products for feeding mice.
We hope it will be helpful.
The major findings are difficult to discern because they are not plainly stated in the text.AU: Thank you for comment. We added some sentences to Results part and we‘ve rewritten the Discussion part.
Has cross-reactivity been shown? There should be a paragraph or two in the Discussion that presents all of the evidence for (or against) the case for cross-reactivity from the experiments and in the context of existing literature.AU: The truly answer is that we did not calculate the cross reactivity and we don’t present it in separate figure. We were thinking about your comment and we agree that there are some digressions in the text but there are no results prove it. We verify that presented experiment was not conducted the way to properly determine the percentage or level of cross-reaction. Competitive ELISA should be done for that, but we have only the titer of specific antibodies determined in indirect ELISA. Finally we decided to change the course of discussion.
It seems like the most interesting aspects of the study are the milk-yogurt comparisons. There should be a paragraph in the Discussion section that summarizes the milk and yogurt differences and what are the possible mechanisms that may be driving these differences. In general, the Discussion section is poorly written...both in content and style. There is a lack of logical flow. There are two sentence paragraphs and incomplete thoughtsAU: We’ve rewritten the Discussion part with hope it will be accepted by Reviewers.
The Discussion should include a paragraph on limitations of the study. Some obvious limitations: use of BSA to test cross-reactivity when it is already a milk protein. Use of a few primers to probe the gut microbiome. What should be studied in the future?AU: However, in order to fully characterize the phenomenon of cross-reactivity protein found only in beef meat of only in milk should be use. The future studies should be focus on possibility induction of immune response by adoptive transfer sensitized lymphocytes, which would be increase knowledge about the phenomenon of cross-reactivity
The Discussion section begins with "This research aimed to show..." Research should never aim to show something. Research should test hypotheses.AU: Thank you for comment. We corrected it.
The authors need to be clear about what fermentation does and does not do. In the Introduction, there is a statement that "lactate fermentation naturally destroys the epitopes of milk proteins". How?? What is the mechanism? What paper demonstrates this? A sentence or two of explanation would help.AU: Ad.1 We agree with this suggestion. The previous text was changed on:
“A similar effect has a lactic fermentation. The reduction of allrgenicty of milk protein during the fermentation process is associated with the destruction of epitopes by hydrolysis of proteolytic enzymes fromLactobacillus. Lactate fermentation naturally destroys the epitopes of milk proteins, which is associated with a reduction in the antigenicity of the fermented products [5,6].”
Abbreviations are used in the abstract before they are defined.AU: The section of abbreviations was added.
All figures are much too small with some text completely unreadable.AU: In all figures the font was changed to a larger one. The quality of graphs was improved by using graphical program. We hope that figures will be readable.
There are spelling and grammar mistakes throughout the manuscript.AU: As I mention earlier we are very sorry for this situation. Before we applied the manuscript to the Nutrients, we ordered a language correction to a professional language service Editage by Cactus. Right now we have no time to correct it by professional. We did some correction by ourselves and we understand that it could be inadequate. We will schedule the service but we will be very grateful if you agree review manuscript in this form.
All figure legends need to explain the parts. If there is an A, B, C, D in a composite figure, then the legend needs to explain what is A. B. C. and D.AU: This mistake was corrected.
All y-axis and x-axis in figures need to be labelled appropriately. For example, Figure 1 y-axis is "concentration". Concentration of what ?AU: We corrected axes. In Figure 2 (earlier 1) the description of axis was changed on: Total IgE concentration [ng/mL]
The units always need to be given. For example, the sentence ending in line 183 is missing the units.AU: We corrected it.
Stylistically, it would be helpful to end every results section with some kind of summary sentence so that the reader knows what is the main finding of that section.AU: The short conclusion was added to every result.
The results shown that humoral response was varied in experimental group. In terminal samples the IgE was the highest in milk group while specific sIgA and IgG was dependent from antigen used in sensitization.AU: The contribution of individual subpopulation of lymphocytes was different in experimental group. The higher contribution of T cells was observed in PBS group both MLN and PP. The yogurt group was characterized percent of Treg in GALT tissue witch was also observed in culture of splenocytes.
The level and profile secreted cytokines in culture medium was different depending s stimulating antigen and experimental group.
The highest level of all tested strains occurred in the yogurt group (exception Bifidobacterium).
Stylistically, it would be helpful to begin each results section with a sentence that either states the hypothesis to be tested or gives the background of why the experiment was done.AU: We try to do this. We hope it will find your acceptance.
The flow cytometry experiments are lacking details. For example, it would be impossible to re-produce Figure 5 from what is presented in the Methods section.AU: We are very sorry for that situation. We modify p.2.7 a little bit to make it more clear.
Paragraph have description of few points of phenotyping;
suspension cells in FACS tubes in constant volume staining cells with the mixture of antibodies to cellular markers washing fixing/permabilization intracellular staining washing and fixing analysis: gating lymphocytes ègating double positive CD3CD4 and CD3CD8 population; from CD3CD4 gating CD25 ègating FoxP3We hope it will clarify results from Figure 5 and 6.
Round 2
Reviewer 1 Report
1. The complete names of the abbreviations should be written and shown in the article when they first appear, and then the acronyms should be presented at all time. 2. The authors should thoroughly go through the sentence structure, spelling and grammar. 3. The experimental design failed to completely encompass and explain the allergies caused by cross-reactivity between proteins indicated. The authors need to elaborate on how the experiments fit with the bigger picture to create a smooth storyline.Author Response
Thank you very much for comments.
We are grateful for all the comments. We have considered all comments while revising our paper. We corrected English language (the certificate is in the zipped file with figures) and we made some changes, which hopefully make the storyline. We are thinking about modify title for “In-VivoStudy of Humoral and Cellular Immune Responses in Mice to β-Lactoglobulin, α-Casein, and Bovine Serum Albumin Allergens in Dairy and Beef Meat” but we were afraid do this because the language correction certificate is for this title (just “Two sources” were change for “Milk and meat”). We have rewritten the text. We change figure one and experiment description in materials and methods part. We hope it will be more clear now.
Comments and Suggestions for Authors
The complete names of the abbreviations should be written and shown in the article when they first appear, and then the acronyms should be presented at all time.AU: We correct all abbreviations and their full meaning. The list is on the end of manuscript. We hope all names were found and exchange in the text.
The authors should thoroughly go through the sentence structure, spelling and grammar.AU: We corrected English language, the certificate is in the zipped file with figures.
The experimental design failed to completely encompass and explain the allergies caused by cross-reactivity between proteins indicated. The authors need to elaborate on how the experiments fit with the bigger picture to create a smooth storyline.AU: We corrected the graphical scheme of experiment (Fig.1). We have rewritten the text trying to logically organize subsequent scientific observations. We hope that we have largely succeeded in improving the clarity of the text.
Reviewer 2 Report
The authors did a nice job of responding to my previous comments and they dramatically improved the paper. There are only a few very minor edits, noted below, which the authors can be trusted to fix.
Line 77: "alergens" should be "allergens"
Line 80: It is still not clear whether this milk is raw or pasteurized. In some countries, both types of milk are available at local markets. The immune response to raw or pasteurized milk would be quite different, so it is an important detail.
Line 204: Tis titers = ? and what is "Ept" ?
Line 312: Recommend changing "demonstrate" to "test". The word "demonstrate" makes the sentence sound as if the authors are trying to show a particular outcome, rather than test a hypothesis.
Line 333: "May be earlier introduction BSA" should be "Maybe earlier introduction of BSA"
Line 334: "need" should be "needs"
Line 349: "Yogurs" should be "Yogurt"
Line 350: "Those" should be "This"
Line 373: "from Yogurt-" should be "from the Yogurt"
Line 393: "TNFand" is missing a space
Line 401: "futer study" should be "future studies"
Line 409: "expose" should be "exposure"
Line 410: "but the needs" should be "but this needs"
Author Response
Thank you very much for your positive assessment of our work. We also thank you for your last comments, which we have improved in the text. By the way, we found a few more letter errors, including "lymphocyytes" on the Fiugre 5A on the OX axis. We've replaced the chart with the correct one.
Line 77: "alergens" should be "allergens"
AU: Thank you we correct it.
Line 80: It is still not clear whether this milk is raw or pasteurized. In some countries, both types of milk are available at local markets. The immune response to raw or pasteurized milk would be quite different, so it is an important detail.
AU: Thank you for this comment we have completed the description. In Poland, milk sold in shops (no matter if it is a local shop or a supermarket) must be pasteurized, so we forgot to specify this parameter.
Line 204: Tis titers = ? and what is "Ept" ?
AU: Thank you for comment. We missed abbreviation in the method description (p.2.7) and abbreviation list. We present titers as Endpoint titer - Ept - what is the reciprocal dilution of the last sample dilution of 0.1 OD above the negative control. We completed the Elisa description (p.2.7) and abbreviations list.
Line 312: Recommend changing "demonstrate" to "test". The word "demonstrate" makes the sentence sound as if the authors are trying to show a particular outcome, rather than test a hypothesis.
AU: Thank you we correct it.
Line 333: "May be earlier introduction BSA" should be "Maybe earlier introduction of BSA"
AU: Thank you we correct it.
Line 334: "need" should be "needs"
AU: Thank you we correct it.
Line 349: "Yogurs" should be "Yogurt"
AU: Thank you we correct it.
Line 350: "Those" should be "This"
AU: Thank you we correct it.
Line 373: "from Yogurt-" should be "from the Yogurt"
AU: Thank you we correct it.
Line 393: "TNFand" is missing a space
AU: Thank you we correct it.
Line 401: "futer study" should be "future studies"
AU: Thank you we correct it.
Line 409: "expose" should be "exposure"
AU: Thank you we correct it.
Line 410: "but the needs" should be "but this ne
AU: Thank you we correct it.